# Adenine-Based Purines and Related Metabolizing Enzymes: Evidence for Their Impact on Tumor Extracellular Vesicle Activities

**DOI:** 10.3390/cells10010188

**Published:** 2021-01-19

**Authors:** Patrizia Di Iorio, Renata Ciccarelli

**Affiliations:** 1Department of Medical, Oral and Biotechnological Sciences, ‘G. D’Annunzio’ University of Chieti-Pescara, 66100 Chieti, Italy; patrizia.diiorio@unich.it; 2Center for Advanced Studies and Technology (CAST), ‘G. D’Annunzio’ University of Chieti-Pescara, 66100 Chieti, Italy

**Keywords:** cancer, extracellular vesicles, exosomes/microvesicles, adenine-based compounds, purine metabolizing enzymes, purinergic receptors

## Abstract

Extracellular vesicles (EVs), mainly classified as small and large EVs according to their size/origin, contribute as multi-signal messengers to intercellular communications in normal/pathological conditions. EVs are now recognized as critical players in cancer processes by promoting transformation, growth, invasion, and drug-resistance of tumor cells thanks to the release of molecules contained inside them (i.e., nucleic acids, lipids and proteins) into the tumor microenvironment (TME). Interestingly, secretion from donor cells and/or uptake of EVs/their content by recipient cells are regulated by extracellular signals present in TME. Among those able to modulate the EV-tumor crosstalk, purines, mainly the adenine-based ones, could be included. Indeed, TME is characterized by high levels of ATP/adenosine and by the presence of enzymes deputed to their turnover. Moreover, ATP/adenosine, interacting with their own receptors, can affect both host and tumor responses. However, studies on whether/how the purinergic system behaves as a modulator of EV biogenesis, release and functions in cancer are still poor. Thus, this review is aimed at collecting data so far obtained to stimulate further research in this regard. Hopefully, new findings on the impact of adenine purines/related enzymes on EV functions may be exploited in tumor management uncovering novel tumor biomarkers and/or druggable targets.

## 1. Introduction

Until a few decades ago, communications among cells were thought to occur essentially by cell-to-cell contacts or via released soluble molecules, able to target structures present in cell membranes (i.e., receptors, ion channels, transporters). More recently, the discovery of extracellular vesicles (EVs), which were initially considered as cargos to transport waste materials [1], has opened a new scenario for signals among cells [2]. EVs are micro/nanostructures that can be released by virtually all cell types as well as detected in body fluids. They usually exchange/transport/donate molecules such as lipids, proteins, nucleic acids and also nutrients, able to implement/alter primitive functions of recipient cells. The half-life of EVs after intravenous injection in nude mice was found to be around 30 min and their clearance was completed within 6 h from injection [3]. Rapid degradation of EVs was also found in human fluids, although precision in this evaluation is still difficult to attain [4]. 

To date, the main difference among EVs is based on their sizes (reviewed by [2,5]), for which they are mostly distinguished into microvesicles (MVs, 0.1–1 µm size) and exosomes (30–100 nm size) [6], although larger EVs have been identified as apoptotic bodies and oncosomes (1–10 µm diameter), the last ones being more cancer-specific. However, the International Society for EVs (ISEV) has recently grouped these particles as small EVs (sEVs), corresponding to exosomes, and large EVs (lEVs) including all other EV types [7]. ISEV has also indicated a series of markers that should be checked to demonstrate the EV nature and the degree of purity of an EV preparation. They include proteins related to Golgi, endoplasmic reticulum, mitochondrial, or nuclear components, which should be present in lEVs, and even more in the large oncosomes, whereas they may be excluded from sEVs (<200 nm) that are presumably formed distant to these locations. Moreover, phospholipids present in lipid bilayers are also potential positive controls for the presence of EVs [7].

In this context, it is important to also consider that the procedures to obtain purified EVs and/or some of their subtypes are various, including ultracentrifugation coupled to size-exclusion chromatography, EV precipitation by commercially available kits or EV capture by antibody-coated magnetic beads, but they are not easily reproducible or insufficient to yield homogeneous material, as reviewed by Gaillard et al. [8]. The same Authors suggest as more specific methods for EV identification the use of antibody-affinity-based methods against major EV proteins such as tetraspanins or, alternatively, sensors against non-denaturable biomolecules such as DNA aptamers, which are single-stranded oligonucleotides with a structure allowing them to bind a specific target. Additionally, there is the possibility to use short protein fragments, which are able to bind major EV membrane proteins or phospholipids. Thus, a more accurate standardization of techniques to isolate and identify EV subtypes is ongoing, which will hopefully allow a better characterization of EVs that represents a critical feature to exploit them for diagnostic and/or therapeutic applications [9]. 

As for the role/function of EVs, this issue is still debated and, thereby, it is intensely investigated, especially in relation to tumors [10,11,12]. Indeed, EV content mostly reflects that of the originating cells and can be involved in the development and progression of malignant cancers by triggering, modulating or suppressing various signal pathways in recipient cells, related to angiogenesis, epithelial–mesenchymal transition (EMT), extracellular matrix (ECM) remodeling, and immune escape [13]. Therefore, unraveling this wide range of activities can disclose new potential clinical applications of EVs in tumors, including diagnosis (as biomarkers via non-invasive liquid biopsy), modulation of their biogenesis and function to limit the transfer of malignancy features, use of EVs to repurpose them as a therapeutic tool in immunotherapy and drug delivery systems. 

Among molecules detected inside EVs, short nonsense nucleic acids, known as microRNAs (miRNAs), which are one of the several types of nucleic acids contained in EVs, and a number of proteins, have already been recognized as able to increase/modulate the aggressiveness of many tumors [14,15]. Very recently, certain attention has begun to be paid to purines, in particular to adenine-based compounds, which have been implicated in tumor growth as extracellular signal molecules [16]. Since also the interaction/relationship between EVs and purines, although at the very beginning, might be relevant to the control of cancer development and progression, in this review we pointed out major findings of this aspect, in order to stimulate research in this direction that can possibly reveal further purine activities in the modulation/control of cancer processes.

### 1.1. Adenine-Based Purines in Brief

Purines are ubiquitous molecules present in all cells from which they are released in the extracellular environment in physiological conditions; however, purine compounds may be also lost by cells as a consequence of plasma membrane damage upon injury [17]. The most studied and functionally characterized among these extracellular substances are the adenine-based purines, mainly adenosine triphosphate (ATP) and adenosine (ADO), which behave as intercellular signals, being able to interact with specific receptors belonging to ADO/P1 and ATP/P2 receptors families [18]. Thus, by activating defined transduction mechanisms, they influence different biological cell/tissue functions ranging from development to proliferation, metabolism, etc. [19,20].

Like inside cells, the extracellular levels of adenine-based purines are finely regulated by ecto-enzymes, which until recently have predominantly been regarded as membrane-bound (ectoenzymes) rather than as soluble or vesicle-associated enzymes. They include, besides a tissue-nonspecific alkaline phosphatase (TNAP), the families of nucleoside triphosphate diphosphohydrolase (E-NTPDase) and nucleotide pyrophosphatase/phosphodiesterase (E-NPPases), which rapidly convert nucleotides (ATP, ADP) into AMP. Subsequently, ecto-5′-nucleotidase (E-5′NT)/CD73 and/or prostatic acid phosphatase (PAP) degrade AMP to ADO. This nucleoside may undergo a further metabolic degradation that converts ADO into inosine and finally to xanthine, by the activity of the membrane-bound adenosine deaminase (ADA) and an extracellular isoform of purine nucleoside phosphorylase (PNP). However, it is to note that nucleosides and derived bases are mostly regained inside the cells by specific transporters, which contribute to clarify pericellular fluid from these substances and to reconstitute the intracellular purine pool. These components (purines, enzymes and transporters) act in an interactive fashion, constituting a complex network called “purinome” [21].

Interestingly, mainly in the last decade, the role of purines in the carcinogenic process has emerged as substances able to modulate tumor development, growth and recurrence. Indeed, ATP and ADO are present in elevated concentrations in the tumor microenvironment (TME). This is in part due to purine leakage through cell membranes likely damaged by hypoxia/inflammation present in the tumor core environment. Moreover, tumor cells are provided with enzymes of the E-NTPDase family, also known as CD39, which, however, slowly transform ATP into AMP that, in contrast, is rapidly degraded to ADO by E-5′NT/CD73. It has also been reported that a number of tumor tissues and cell lines express purine receptors belonging to the P1 and P2 receptor families, which are prevailingly activated by their natural ligands that are ATP or ADO, or by synthetic analogs, leading to a variety of effects ranging from tumor cells proliferation to increase in cell migration and aggressiveness, modification of tumor cell metabolism and ability of immune attack evasion (reviewed by [22]).

### 1.2. Purines and EVs Derived from Normal or Pathological Non-Cancer Cells/Tissues

As aforementioned, possible relationships between extracellular purines and EVs have so far been poorly investigated. The first evidence of this was published in 1983 by Kanabe et al. [23], who investigated the localization of ATP-hydrolyzing activity on membrane-bound extracellular matrix vesicles. These particles were previously identified as derived from the membrane of chondrocytes, osteoblasts and odontoblasts by a process of cortical budding [24] and are involved in the initiation of hard tissue calcification. In this context, Kanabe et al. [23] demonstrated by studies of electron microscopy that ATPase activity was present on both the outer and the inner surface of the membranes of those vesicles. The authors suggested that the activity of this enzyme could serve to release orthophosphate in the matrix, which is necessary to form the first crystals of hydroxyapatite and thereby to start calcification. 

After several years from those pioneering researches, some studies reported findings obtained by a variety of experimental models and conditions, which are summarized in Table 1, in which a brief mention is given also on the types of analyses performed for the identification of purine compounds and enzymes in EVs. One of these research studies showed that ATP stimulation of P2X7 receptors, present on a monocyte/macrophage-like cell line, THP-1 cells, caused a prompt release of interleukin 1β (IL1β) via MVs [25]. Later on, these results were confirmed by data demonstrating that stimulation of the same receptors in microglia by exogenous ATP or ATP released from co-cultured astrocytes induced the release of EVs containing IL1β [26]. Subsequently, it was suggested that ATP release from astrocytes could occur in a vesicular form [27]. In the following decade, it was reported that cells, such as endothelial cells, astrocytes and pericytes forming the blood–brain barrier (BBB), shed MVs expressing NTPDase after oxygen–glucose deprivation, as evaluated by reverse transcriptase-polymerase chain reaction (RT-PCR) assay. This enzyme, by increasing the breakdown of ATP released to toxic levels, could have a protective/modulatory effect on all brain parenchyma [28]. Further findings showed that astrocytes shed large membrane vesicles containing lipid droplets and ATP [29]. Interestingly, ATP, identified by quinacrine staining, was associated with mitochondria released into vesicles and visualized by tetramethylrhodamine methyl ester perchlorate (TMRM).

Since then, many other reports have demonstrated that astrocytes release different EV types containing a variety of substances, for which they are considered as the main secretory cells of the central nervous system [30]. However, apart from sporadic studies, the interest in purines as modulators of EV production/activity or as compounds contributing to EV equipment has shown a decrease over the past 5–6 years, re-emerging in a number of papers published within the last 2 years.

Thus, it has very recently been reported the ability of different cells from normal tissues/fluids to release EVs containing purine metabolizing enzymes. This is the case of EVs isolated from rat submandibular gland or saliva, in which enzymes able to hydrolyze ATP up to ADO were identified by immune-based assays and biochemical analysis [31]. Based on preliminary results, the same authors affirm that also EVs isolated from human saliva contain the same metabolic chain. Since an increase in extracellular ATP (eATP) levels has been reported in periodontal disease, leading to alveolar bone resorption and bone loss [32], the results above described suggest that the control of eATP, also by EV enzymes, might contribute to significantly reduce bone loss in periodontitis. In line with these findings, Gonzales and coll. [33] have also shown that histamine, which is present in pro-inflammatory environment such as that of periodontal disease, can stimulate the secretion of EVs with ecto-nucleotidase activity from the rat submandibular gland. In this way, histamine, by activating its H_4_ receptors, may contribute toward regulating the pro-inflammatory eATP levels, favoring the formation of tissue-protecting ADO. Again, remaining in the field of Dentistry, it is noteworthy that human mesenchymal stem cells (MSCs), which are of potential therapeutic use in bone defects, are able to release exosomes, which increased in vitro migration and proliferation of cells from periodontal ligament through ADO formed by EV-derived CD73 activity [34]. Interestingly, the same exosomes, loaded on a collagen sponge and administered in an immunocompetent rat model, enhanced the regeneration of surgically-induced periodontal bone defects.

Noteworthy, only a few papers have reported that ATP is contained in EVs. One of these showed that ATP can be also produced in exosomes derived from seminal plasma and it normally assures a safe environment for sperm viability and fertilization in mammals. For instance, such ATP regulates boar sperm motility [35]. In agreement with these data, a recent review collected findings on the possibility that EVs are able to support surrounding cells by glycolytic formation of ATP [36]. The author in particular emphasizes that prostate epithelial cells form exosomes, called prostasomes, and secrete them into semen, with the main task to adhere to sperm promoting its motility to reach eggs [37]. These prostasomes can produce ATP by glycolysis of fructose, available in semen, and the produced ATP is rapidly consumed by adjacent prostasomal ATPases, contributing toward maintaining the important energy-consuming sperm functions. 

Furthermore, the effect of purine receptors on EV production/release has also been scarcely investigated. It was recently reported that platelets activated by ADP release EVs (PEVs), which are characterized as PEVs^-^, when positive for platelet glycoproteins IIIa (CD61) and negative for P-selectin, and as PEVs^+^, when positive for both markers. The contemporaneous exposure of platelets to antagonists of P2Y1 (P2Y1R) and P2Y12 (P2Y12R) receptors, expressed on their plasma membranes, reduced PEVs^-^ release, whereas the antagonists alone were ineffective. In contrast, PEV^+^ release from platelets was reduced only by P2Y12R antagonist, administered alone, or by combined exposure to the two antagonists. Noteworthy, both types of PEVs triggered coagulation when tissue factor was also present. Thus, the observed clinical effects of the anti-platelet drug ticagrelor, a P2Y12R antagonist, may be due to its ability to inhibit platelet aggregation and also decrease PEV release from activated platelets [38]. 

Other findings demonstrated that EVs may act as a compartment to store second messengers such as cyclic adenosine monophosphate (cAMP) [39]. In this study, EVs were isolated from cultured endothelial cells of different vascular beds, being those isolated from pulmonary microvascular endothelial cells (PMVECs) able to release the highest EV amount. Interestingly, exposure of these cells to agents increasing near-membrane cAMP levels also enhanced cyclic nucleotide formation, assayed by enzyme immunoassay and quantified by high-performance liquid chromatography (HPLC) coupled to mass spectrometry (MS) within the corresponding EVs, while no increase in EV number was observed. Noteworthy, EVs containing elevated cAMP, added to confluent PMVEC cultures, maintained cell monolayer resistance. To confirm data obtained from the cell cultures aforementioned, the Authors also investigated changes of cAMP levels in EVs isolated from the perfusate of rat lungs, uncovering that they were augmented when cells were stimulated with agents able to stimulate cAMP formation such as isoproterenol and rolipram. From these data it is possible to hypothesize that cAMP released from EVs could contribute in vivo to protect the pulmonary endothelial barrier, avoiding the formation of pulmonary edema [40].

In contrast, recent literature is richer in reports showing the presence of purines inside EVs or their formation through metabolic enzymes found in EVs as well as their influence on EV function in inflammatory conditions. For example, in agreement with the findings reported above, Kriebel et al. [41], using as experimental model cultures of amoeba Dictyostelium discoideum, which are able to send chemotactic signals among neighboring cells, demonstrated that MVs released from these cultures not only contain cAMP, but also actively synthetize and release cAMP to promote chemotaxis. Interestingly, to detect cAMP levels, they used a competitive cAMP assay based on time-resolved fluorescence resonance energy transfer (FRET), which has higher sensitivity than conventional fluorescence-based assays. Moreover, the Authors showed through different experimental approaches that cAMP is released from MVs via a specific ATP-binding cassette transporter, ABCC8. Further findings from another research group [42] using experimental models in vitro and in vivo, demonstrated that extracellular ATP promotes the release of MVs from macrophages, which, besides IL1β, also contain tumor necrosis factor (TNF, best known as TNFα), mainly as transmembrane precursor (pro-TNF, 26 kDa) rather than as soluble form. ATP also induced EV secretion, favoring the release of TNF precursor, which targets cells more effectively than soluble TNF, and causing significant TNF-dependent inflammation. These data might explain why anti-TNF therapies against soluble TNF are poorly useful in acute inflammatory diseases [43]. A final example comes from the review by Schneider et al. [44], who collected papers, some of which deal with the presence/expression of 5′-ectonucleotidases in EVs. In particular, these enzymes are usually distinguished into a cell membrane-bound enzyme, also known as CD73, and a soluble isoform, called AMPase. Both have been detected in exosomes released from regulatory T (Treg) lymphocytes [45], MSCs [46], as well as in EVs isolated from human plasma [47], mainly by the use of immunofluorescent antibodies coupled to flow cytometry analysis. From the examination of the current literature, the authors argued that the enzymes responsible for the generation of ADO from AMP are readily shed from cells by this novel way and can be transported to act on other cell types, thus widening their range of action at the site of inflammation and implementing the activity of ADO as a regulator of the immune response [48,49]. 

Of note, this last aspect has been mostly evaluated in tumors, in particular as concerns the activity/function of EVs, whose presence has been identified in the TME of different cancers [50,51], as discussed below and recently reviewed by Azambuja et al. [52].

### 1.3. Focus on Adenine Purines and EVs in Cancer

Some years ago, Clayton et al. [50] showed that small vesicles (exosomes) secreted by diverse cancer cells can contribute to the extracellular ADO production. Indeed, they found that those particles exhibited ATP- and 5′AMP-phosphohydrolytic activity, due to the expression of CD39 and CD73, respectively, which were identified by Western blot analysis. Similar enzyme activity was also detected in exosomes obtained from pleural fluid of mesothelioma patients, which led to the formation of ADO, in turn contributing to negative regulation of T cell function. The authors concluded that such a mechanism might play a wider role in reducing immune responses often observed in cancer. 

Since then, a number of research groups have investigated the interactions between EVs and the turnover of adenine-based purines in cancer, as summarized in Table 2, wherein some details of the methods used to detect purines and related enzymes in EVs are reported. Thus, ADO forming enzymes were detected in exosomes released from multiple myeloma (MM) cells by flow cytometric analysis. They belong to either the canonical pathway, which includes NTDPases/CD39 and CD73 working in tandem to produce ADO, or the noncanonical pathway, which generates ADO via other enzymes. These are: nicotinamide adenine dinucleotide (NAD^+^)-glycohydrolase/CD38, which converts NAD^+^ to ADP-ribose (ADPR), and CD203a (PC-1), degrading ADPR to AMP that is subsequently metabolized by CD73 to ADO. Thus, both pathways have CD73 as a common link [53]. Additionally, myeloma-derived exosomes mediated ADO increase also upregulating the expression of ADO generating enzymes in recipient cells [54]. Interestingly, levels of ADO in plasma samples from the bone marrow of MM patients were higher than those measured in patients with other hematological malignancies and correlated with the disease stage [55]. Therefore, these results would indicate that ADO production by the activity of purinergic enzymes present in exosomes from MM cells may contribute to the disease severity. Further data have confirmed the immunomodulatory role of adenine purines and related metabolizing enzymes contained in cancer-derived EVs, even though data are in part conflicting.

For example, it has very recently showed that glioma cells release EVs [56]. These particles were isolated from rat C6 glioma cell supernatant and characterized by size, expression of some markers including CD39 and CD73 and ability of producing ADO, identified by a selective chromatographic analysis. EVs were also injected into the right striatum of Wistar rats together with C6 cells (to reproduce the tumor in vivo) or intranasally administered to rats bearing C6 cells. Analysis of the results showed a reduction in tumor size and cell proliferation. Thus, the Authors conclude that EVs administered before or at early step of tumor growth might have antiproliferative properties. Moreover, EV administration in vivo caused a reduced expression of rat immune cell marker FoxP3, which, when present in excess in T cells, could inhibits lymphocyte ability to restrain the formation of cancerous cells. 

Different results and conclusions were obtained by evaluating the expression levels of genes related to adenine purine metabolism in tissues of patients with head and neck squamous cell carcinoma (HNSCC), which were upregulated as compared to normal tissues. Likewise, a high number of purine metabolites, mainly ADO and inosine, were determined in exosomes from plasma of the same patients or from supernatants of UMSCC47 cells, an HNSCC cell line, using ultraperformance liquid chromatography (UPLC). Noteworthy, adenine purine levels were significantly increased in exosomes from patients at early-stage disease without lymph node metastasis, and decreased in those from patients at more advanced cancer stage. Thus, modifications in the content/levels of adenine purines in circulating exosomes may be a prognostic index for the clinic HNSCC progression, since the observed increase in the exosome transport of ADO, which may cause immunosuppressive effects, is possibly involved in cancer progression [57]. Other findings would also suggest ADO-induced immunosuppression via EV contribution. Thus, Tadokoro et al. [58] showed that EVs secreted by cell lines of breast cancer are disrupted by perforin, a pore-forming protein liberated by CD8^+^ cytotoxic lymphocytes (CTLs), when activated by the known cytokine interferon-gamma (IFN-γ), whose release in vivo is generally stimulated by the tumor presence and aims at rejecting the tumor itself [59]. The EV destruction by perforin may liberate ADO in the TME, as detected by a sophisticated procedure coupling liquid chromatography to mass spectrometry analysis (LC-MS), which in turn hinders perforin secretion by CTLs. Since adenosine deaminase, which degrades ADO, reversed the decrease in perforin extrusion from CTLs, these results confirm the role of ADO as an immunosuppressive agent, likely by interaction with A2A receptors on the CTL membrane [50]. They also suggest a novel pathway for cancer survival via cell-derived EVs. Noteworthy, immunosuppression caused by impairment of CLT function may greatly affect the tumor response to chemotherapy. Thus, Zhang et al. [60] demonstrated that EVs released from rodent B lymphocytes and not those from NIH-3T3, which are a line of fibroblast-like cells, contain high levels of CD39 and CD73 that hydrolyze ATP released from chemotherapy-treated tumor cells into ADO, in turn inhibiting CTLs and therefore attenuating the chemotherapeutic efficacy. Data were extended to patients with different cancers, in whose serum EVs were found to be similar to those from rodents, since they expressed high levels of CD39 and CD73, which effectively converted ATP into ADO. Moreover, the presence of those EVs in patient serum before chemotherapy negatively correlated with their free-survival period and cancer progression. In line with the importance of immune surveillance are also further recent findings. Gorzalczany et al. [61] have shown that mast cells (MCs), present in TME and exerting a pro-tumorigenic role, may be activated and yet reprogrammed by contact with neighbor tumor cells or by EVs released by these latter. More in detail, they demonstrated that EVs derived from pancreatic and lung cancer cell lines stimulate MCs by CD73-mediated formation of ADO. This was coupled to activation of ADO/A3 receptors, which contributed to upregulation of some angiogenic genes such as IL8 and IL6, vascular endothelial growth factors (VEGF) and amphiregulin, a ligand of epidermal growth factor receptors (EGFRs) playing an important role in lung cancer progression. In contrast, Angioni and coll. [62], who recently investigated the properties of EVs released from murine bone marrow MSCs in response to pro-inflammatory cytokines, reported that these EVs, enriched in CD39 and mainly CD73, as identified by Western blot analysis, inhibited endothelial cell migration in vitro and angiogenesis in a murine model of cancer in vivo. The anti-angiogenic mechanism was ascribed to EV-induced ADO production, which, via ADO/A2B receptors, caused a NADPH oxidase 2 (NOX2)-dependent oxidative stress within endothelial cells. Since MSCs may regulate vessel remodeling, these results could open a new perspective in the use of MSC-derived EVs for anti-angiogenic therapies in tumors. 

From the data reported above, it is clear that the opposite results obtained in relation to ADO activity on the immune system may be due to the adoption of different experimental models/animal species. However, most data agree on the fact that tumor progression/relapse is often associated with defective immune-surveillance. In this context, an important role could be played also by natural killer (NK) lymphocytes [63]. At present, it is not possible to rule out that ADO, deriving from eATP degradation also thanks to the activity of enzymes contained in EVs, may suppress the maturation of these cells. Indeed, Neo and coll. [64] reported that tumor-infiltrating NK cells upregulate CD73 expression upon contact with tumor cells and the frequency of these CD73-positive NK cells correlated with larger tumor size in breast cancer patients. Their findings showed that NK cells transport CD73 to the cell surface and release it into the extracellular space via an exocytotic process. Consequent to these events, CD73-positive NK cells acquired nontypical functions to suppress the immune environment, thus favoring tumor escape from immune control. Thus, future studies might explore whether NK lymphocytes may exert a more general immune-suppressive function in tumors, as a consequence of CD73 activity acquired via EVs from cancer cells, many of which express this enzyme at high levels [65].

## 2. Discussion and Conclusions

From the findings above collected it is evident that research is moving also to uncover the contribution of purines to the EV activities/functions in cancer. 

One important aspect emerging from literature is the general convergence on the fact that EVs, in particular exosomes, which are the most investigated and best characterized (nano)particles released from cells, are rich in enzymes degrading purines, with a major emphasis pointed on CD39 and CD37. This finding is not surprising considering the modality of formation of EVs, which derive from cells including parts of their plasma membranes. Indeed, MVs bud directly from the plasma membrane, while exosomes are generated within inner multivesicular bodies that, upon maturation, fuse with the plasma membrane. Since CD39 and CD37 are anchored to cell membrane phospholipids and located in particular zones adjacent to caveolae [66,67], it can be expected that EVs, more likely their membranes, are provided with these enzymes. 

Also relevant is that these enzymes have often been recognized as features correlated with cancer malignancy [65]. This is compatible with the fact that cancer initiation, as well as progression/relapse, are processes also ascribed to the presence of cancer stem-like cells (CSCs) inside the tumor mass [68], which are provided with self-renewal potential, high migration/invasion ability and resistance to anti-cancer drugs. Interestingly, purinergic enzymes, in particular CD73, are considered as a distinctive sign of stemness in normal MSCs [69] and also of malignancy in a number of CSCs from different tumors [70,71,72]. Thus, it could be expected that EVs deriving from tumors, which comprise also these more malignant cells, are characterized by high expression levels of purine enzymes. Likely, to exploit these data at best, it should be more systematically explored the presence of EVs with high expression levels of CD73 in human body fluids, evaluating its possible relationship with cancer malignancy degree or tumor progression. If so, CD73 could be a suitable candidate in oncology for diagnostic purposes, at least.

However, aside from the obviousness of the presence of purine enzymes on/in EVs, why do they have to function in addition to the same cell enzymes? As noted above, ATP is detected in TME at high concentrations [73]. These high and persistent levels could mine tumor cell survival due to ATP interaction with different receptors, whereas lower and short-lasting ATP concentrations would be of support to cancer progression [74]. In contrast, ADO, deriving from eATP thanks to the tandem activities of CD39 and CD73 expressed by cell/EV membranes and formed at high levels in TME as well, would help tumors in their growth/expansion, mainly by controlling the efficiency of the immune apparatus. On this aspect, there is a wide consensus. Indeed, ADO, via the interactions with P1 receptors [75], exhibits an indirect “protective” role on tumors by affecting the function of almost all cells of the immune system, including MCs and B, CD8^+^ CT, NK [50,58,60,64], and also Treg lymphocytes, of which ADO promotes the immune-suppressive activity [47]. Since it is even more evident that chronic inflammation may contribute to about 25% of human cancers [76], the overall behavior of TME/EV purines and enzymes is similar to that of the same compounds in inflammatory conditions [77]. Indeed, in inflammation ATP and related nucleotides are initially found at high concentrations and, activating P2 signaling, they promote pro-inflammatory immune response to circumscribe the affected area and fight the inflammatory agents. Over time, nucleotides are metabolized to ADO, which exerts anti-inflammatory functions. Thus, in both conditions, it can be hypothesized that the activity of purine enzymes in EVs would corroborate that of the inflamed/tumor cells in defending tissues/tumors from extracellular aggressive factors. Moreover, it cannot be ruled out that purine enzymes might be donated via EVs to neighboring normal/tumor cells, thus allowing them to modulate the presence/formation of purines and their consequent direct/indirect activities sustaining normal tissue/tumor survival. Clearly, the final outcome of the activity of purines and related enzymes in inflammation or tumors is different in that the purine system is aimed at controlling inflammation favoring the return to a normal healthy condition, whereas in tumors the activity of the purinergic network seems to mainly favor tumor growth/maintenance. 

In addition to data here included and discussed, in our opinion investigation on possible connections between purines and cancer-derived EVs should be enlarged to other aspects, which are still scarcely investigated or lacking. One of these might be the research on the possible existence of an asset of adenine-based purines in EVs. Indeed, it is well known that uncontrolled cell proliferation is a hallmark of cancers and that purines and enzymes for their de novo synthesis, which are fundamental/necessary for cell proliferation, are increased in tumor cells as well as the energy expenditure [9]. Thus, like for normal cells, EVs, possibly released from more quiescent tumor cells or peritumoral tissue, might transport purines to be relocated in more needy tumor cells as precursors of nucleic acids or as a source for energy support.

But, more importantly, it should be better explored whether activation/inhibition of purine receptors expressed by tumor cells, besides producing pro-/anti-tumor effects by directly acting on molecular pathways within the same cells, may also control the EV formation/release from them. This could be an additive or alternative way to modulate the survival/death of the tumor itself, if EVs are vehicles to supply tumors with resistance factors. In light of the current literature, it is possible to speculate that purine receptors, in particular those belonging to the P2 family, could stimulate the formation/release of EVs. Indeed, in EV making/secretion mechanisms there is a rearrangement of the plasma membrane of the cells from which EVs originate [2]. Since P2 receptors, and in particular, the ionotropic P2X7 receptors, are able to influence membrane plasticity [78,79], then, control of purinergic signaling may lead to modulate also EV formation/release, thus providing possible beneficial anti-tumor effects. Of course, this is only an example and there may be other possibilities in terms of receptor signals/mechanisms that should be evaluated/investigated.

In future investigations, it would be of crucial importance that the characterization of EVs is performed following the indications of the ISEV, also as for EV nomenclature. As well, the identification of purines compounds in TME/EVs and the evaluation of the activity of the related metabolizing enzymes need to be performed by standardized and advanced methods. Following these criteria will allow the findings can result in more homogenously interpretable and more efficiently used for diagnostic/therapeutic purposes.

In conclusion, given the great attention today paid to the properties/functions of EVs in cancer as well as the emergent role played by purines as substances able to affect aggressiveness/progression of different tumors, we think that implementation of studies on the possible relation between tumor-derived EVs and the purinergic system would lead toward discovering new potentialities of this machinery, which could be suitably modulated and hopefully exploited for adjunctive/innovative as well as diagnostic/therapeutic cancer management.

## Figures and Tables

**Table 1 cells-10-00188-t001:** Evidence for the presence/expression of purinergic system components in extracellular vesicle (EV) released from different noncancer cells.

EV Type	Source	Relationship with the Purine System	Function	Techniques Used for Detection of Purines and Related Enzymes	Ref.
EVs	Membrane of chondrocytes, osteoblasts and odontoblasts	Presence of ATP-hydrolyzing activity	Release of orthophosphate in the matrix initiation of hard tissue calcification	Electron microscopic cytochemical methods	[23]
MVs	A monocyte/macrophage-like cell line, THP-1 cells	ATP stimulation of P2X7 receptors present on THP-1 cells	IL1β release	None	[25]
EVs	Microglia	ATP exogenously added or released from co-cultured astrocytes	IL1β release	None	[26]
MVs	Endothelial cells, astrocytes and pericytes, forming the blood–brain barrier (BBB)	Presence of NTPDase activity after oxygen–glucose deprivation	Modulation of BBB to brain ischemic events	Enzyme detection by RT-PCR analysis. Enzyme activity assayed by cytochemistry	[28]
Large vesicles	Astrocytes	Mitochondria, lipid droplets and ATP	Sending signals to neighboring neural cells	Vesicles labeling by quinacrine to stain ATP linked to TMRM-loaded mitochondria	[29]
EVs	Rat submandibular gland or saliva (likely also in humans)	Contain enzymes (NTPDAses and E-5′NT) deputed to hydrolyze ATP up to ADO	Possible contribution to reduce eATP in periodontal disease	Enzyme activity assayed by biochemical methods. Enzyme localization by Western blot and immuno-TEM analyses	[31]
Exosomes	MSCs	CD73 activity	Formation of ADO increasing migration and proliferation of periodontal ligament-derived cells	Indirect evidence for ADO formation/activity based on the use of P1 receptor antagonists	[34]
Exosomes	Seminal plasma	Glycolytic formation of ATP	Promotion of sperm motility	Enzyme activity evaluation by biochemical assay	[35]
Prostasomes	Prostate epithelial cells	Glycolytic formation of ATP	Sperm functions	Enzyme activity evaluation by biochemical assay	[37]
EVs	Platelets	Expression of P2Y1R and P2Y12R	Triggering coagulation when tissue factor was also present	None. Indirect evidence of the wo receptors by the use of related antagonists.	[38]
EVs	Endothelial cells from different vascular	Ability to form cAMP protecting the pulmonary endothelial barrier	Maintenance of monolayer resistance when cells were cultured in vitro	cAMP formation assayed by enzyme immunoassay and quantified by HPLC-MS	[39,40]
MVs	Amoeba Dictyostelium discoideum (a model to study inflammation)	Ability to form and release cAMP	Promotion of chemotaxis	cAMP formation measured by time-resolved FRET	[41]
MVs	Macrophages	Stimulation by ATP causes release of TNF precursor	TNF-induced inflammatory condition	None	[42]
Exosomes	Different sources (Treg lymphocytes, MSCs, human plasma)	Membrane-bound CD73 and soluble enzyme isoform, AMPase	Ability to form ADO from extracellular AMP to implement ADO activity as regulator of immune response	Enzyme expression by immuno-fluorescence staining; ADO production analyzed by HPLC	[44,45,46,47]

**Table 2 cells-10-00188-t002:** Evidence for the presence/expression of purinergic system components in EV released from different cancer cells.

EV Type	Source	Relationship with Purine System	Function	Techniques Used for Detection of Purines and Related Enzymes	Ref.
Exosomes	Different cancer cells	Expression of CD39 and CD73	Formation of ADO, in turn contributing to negative regulation of T lymphocyte function	CD39 identified by Western blot analysis; its activity measured by a luciferase-based ATP assay. CD73 identified by a fluorescent antibody; its activity measured by a colorimetric assay to evaluate inorganic phosphate production by exogenously added 5′AMP.	[50]
Exosomes	Multiple myeloma cells	Expression of purine degrading enzymes of the canonical (CD39/CD73) and noncanonical pathways (CD38/CD203a) to form ADO	ADO-induced up-regulation of the same enzymes also in recipient cells. ADO levels correlated with disease severity	Enzymes identified by fluorescent antibodies and flow cytometric analysis. Purine compounds identified by HPLC analysis.	[53,54,55]
EVs	Rat C6 glioma cells	Expression of CD39/CD73 with ADO formation	In animal models of tumor in vivo EV administration reduced tumor size	Enzymes identified by fluorescent antibodies and flow cytometric analysis. Purine compounds identified by reversed phase ion-pair chromatography	[56]
Exosomes	UMSCC47, a head and neck squamous cell carcinoma (HNSCC) cell line, or plasma of patients with HNSCC	Increased adenine purine levels in exosomes	Possible ADO immuno-suppressive effects associated to cancer progression	Purine enzymes analyzed by gene expression levels using the Cancer Genome Atlas (TCGA) database. Purine levels quantified by UPLC.	[57]
EVs	Cell lines of breast cancer	EVs disrupted by perforin liberated from IFN-γ-activated CD8^+^ cytotoxic lymphocytes (CTLs) release ADO	ADO hinders perforin secretion by CTLs, confirming to be an immunosuppressive agent	Purine compounds identified by LC-MS.	[58]
EVs	Rodent B lymphocytes and serum from patients with different cancers	High levels of CD39 and CD73	ATP hydrolysis to ADO that inhibits CTL anti-tumor activity	CD39 and CD73 expression in EV was detected by flow cytometry. ATP degradation and adenosine formation evaluated by a commercial kit	[60]
EVs	Pancreatic and lung cancer cell lines	Stimulation of macrophages by CD73-mediated ADO formation	ADO via A3 receptor contributed to upregulation of different factors (IL8 and IL6, VEGF and amphiregulin, EGFR ligand) playing an important role in cancer progression	Indirect evidence for ADO formation/activity by the use of selective A3R antagonists	[61]
EVs	Murine bone marrow MSCs	High levels of CD39 and CD73 that induced ADO production	Inhibition of endothelial cell migration in vitro and angiogenesis in a murine model of cancer in vivo.	Enzymes identified by Western blot analysis; ADO production quantified by a commercial kit	[62]
Vesicles	Natural killer lymphocytes	CD73 release	Immune-suppressive function favoring tumor escape	CD73 detected by a specific ELISA method commercially available. Its activity evaluated through relief of AMP-mediated inhibition of ATP detection in a luciferase-based system.	[64]

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
