# Peer review of "Adenine-Based Purines and Related Metabolizing Enzymes: Evidence for Their Impact on Tumor Extracellular Vesicle Activities"

_cells, 2021, doi:10.3390/cells10010188_

Round 1
Reviewer 1 Report
Dear authors,
Your review is a good description about the potential of purine metabolism in cancer to develop new therapies but I have some suggestions to improve the manuscript.
Comment 1. The authors should review the nomenclature of extracellular vesicles following the rules of ISEV in the manuscript because exosomes and microvesicles are not used in the EV field, the researcher use large extracellular vesicles (lEV) or small extracellular vesicles (sEV). Between 41-47, they must mention the EV markers.
Comment 2. In the lines 57-58, they should reference other nucleic acids contained in EV like long-coding RNA, DNA...
Comment 3. In the line 96, the authors should explain which natural ligands are and the references
Comment 4. In the table 1 and 2, the authors must change the nomenclature of EV
Comment 5. In the table 1, in the reference of 26, the authors must put the name of enzymes associated with hydrolyze ATP up to ADO
Comment 6. The authors have missing the abbreviations (CD61 and P2Y12)
Comment 7. In the lines 193 to 199, the authors must explain which type TNF is mentioned (alpha, beta...)
Reviewer 2 Report
Doctors Di Iorio and Ciccarelli presented a very interesting and well structured paper, critically reviewing the knowledge about a particular aspect of EVs cargo, the adenine-based purine and their related enzymes. The Review is well written and quite complete. Just some suggestions:
- The paragraph from line 48 to 56 lacks of some references. I suggest PMID 32878063, PMID 33316884 and PMID 32216070.
- In the introduction, the Authors refer to isolation techniques and EVs sub-types identification standardization. The recent investigation of tumor-derived exosomes and EVs enrichment, based on either peptide affinity or immuno-capture, should be introduced.It has been explored both in solid tumor and in leukemia. Moreover, it could improve the selection of EVs carrying adenine-based purines and help the investigation of this interesting aspect in oncology, maybe by new EVs membrane markers selection (e.g. CD73, as Authors stressed?).
- Line 228. MM: do the Authors refer to Multiple Myeloma? If yes, please specify it in the previous sentence.
- I strongly suggest the Authors to improve the Review with a Figure or a Table, as well as a brief paragraph or chapter, concerning the available techniques to study the presence of vesicular adenine-based purines and related enzymes. The Authors well presented the surprising results and evidences obtained by the commented studies, given details about the in vitro and in vivo models. Nevertheless, few is reported about the methods: Western blotting? Mass spectrometry? Immuno-binding? An overview of the available or future approaches will be appreciated.
- I am wondering about Authors' conclusions and hypothesis about non-tumor diseases. Maybe some suggestions or future perspectives should be add to Discussion and conclusion section.
- There are some typo and space errors here and there.
I hope my suggestions will help the Authors in improving the manuscript.
Author Response
Pleas see the attachment
